# Aldosterone Increases Vascular Permeability in Rat Skin

**DOI:** 10.3390/cells11172707

**Published:** 2022-08-30

**Authors:** Michal Aleksiejczuk, Anna Gromotowicz-Poplawska, Natalia Marcinczyk, Joanna Stelmaszewska, Janusz Dzieciol, Ewa Chabielska

**Affiliations:** 1Department of Biopharmacy, Medical University of Bialystok, Mickiewicza 2C str., 15-222 Bialystok, Poland; 2Department of Pharmaceutical and Biopharmaceutical Analysis, Medical University of Bialystok, Mickiewicza 2D str., 15-222 Bialystok, Poland; 3Department of Human Anatomy, Medical University of Bialystok, Mickiewicza 2A str., 15-230 Bialystok, Poland

**Keywords:** aldosterone, vascular permeability, skin microcirculation, eplerenone, von Willebrand factor

## Abstract

The aim of this study was to evaluate the effect of acute aldosterone (ALDO) administration on the vascular permeability of skin. ALDO was injected intradermally into rats, and vascular permeability was measured. Eplerenone (EPL), a selective mineralocorticoid receptor (MR) antagonist, was used. Skin biopsies were carried out for immunohistochemical (IHC) staining, and polymerase chain reactions were performed to analyze the expression of MR, 11β-hydroxysteroid dehydrogenase type 2, von Willebrand factor (vWF), vascular endothelial growth factor (VEGF), and zonula occludens 1. Our study showed the presence of MR in the rat skin vasculature for the first time. It was found that ALDO injection resulted in a more than 30% increase in vascular permeability and enhanced the endothelial exocytosis of vWF. The effect of ALDO diminished after EPL administration. An accumulation of vWF and a reduction in VEGF IHC staining were observed following chronic EPL administration. No effect of ALDO or EPL on the mRNA expression of the studied genes or skin structure was observed. The results suggest that ALDO increases vascular permeability in the skin via an MR-dependent mechanism. This effect of ALDO on skin microcirculation may have important therapeutic implications for diseases characterized by increased levels of ALDO and coexisting skin microangiopathy.

## 1. Introduction

Aldosterone (ALDO) is the end product of the renin–angiotensin–aldosterone system. It regulates renal sodium reabsorption, as well as the blood volume, via the activation of the mineralocorticoid receptor (MR) in the kidneys. ALDO is synthesized de novo not only in the adrenal cortex, but also in other tissues, such as the heart, blood vessels, brain, and skin [1,2,3,4,5]. So far, the function of ALDO and MR in skin microcirculation is unclear, and the role of ALDO in skin physiology and pathology remains unknown.

In mouse models, constitutive MR knockout has been found to lead to early postnatal death, apparently due to uncontrolled renal salt loss, whereas the inducible overexpression of skin MR resulted in developmental and postnatal abnormalities of the epidermis and hair follicles [6]. Cutaneous abnormalities, such as epidermal atrophy and progressive alopecia, were observed in mice overexpressing MR [7]. To the best of our knowledge, no study has analyzed in detail the functional role of MR in the dermis and skin vasculature. Taking into account that extrarenal MR is involved in pathological action in the heart and blood vessels (e.g., inflammation, vasculopathy, and endothelial dysfunction) rather than in physiological homeostasis, it could also be proposed that skin MR is responsible for the deleterious effects of ALDO in inflammation-related skin diseases and skin vascular disorders [8]. Histopathological alterations, such as epidermal hyperplasia, impaired differentiation, and increased dermal infiltrates, were noted in skin samples from primary aldosteronism (PA) patients, which correlates with increased NF-κB signaling and upregulation of TNF-α and IL-6 cytokines. These skin samples also showed a higher expression of MR, the glucocorticoid receptor (GR), and 11β-hydroxysteroid dehydrogenase type 2 (HSD11β2) [9]. In addition, altered skin perfusion, negatively correlated with ALDO plasma levels, and greater cutaneous microvascular dysfunction, were observed in patients with PA, compared with patients with essential hypertension [10]. Thus, based on these findings, it could be suggested that ALDO plays a pathophysiological role in skin microcirculation.

Endothelium forms a semipermeable barrier that dynamically regulates the transport between blood and tissue [11]. Under physiological conditions, the endothelial monolayer in the skin vasculature is impermeable to molecules bigger than 40 kDa [12]. Intracellular junction proteins, such as tight junctions and adherens junctions, provide vascular integrity [13]. ALDO exerts a pathological impact on the endothelium, affecting its function [1,14]. Several mechanisms have been shown to contribute to ALDO-induced endothelial dysfunction, such as ALDO-mediated vascular tone dysfunction, ALDO- and endothelium-mediated vascular inflammation, ALDO-related atherosclerosis, and vascular remodeling. ALDO affects vascular inflammation by activating MR, leading to the transcription of proinflammatory genes (IL-1β, IL-6, CTLA-4, and PAI-1) [5,15,16]. Moreover, ALDO can promote inflammation via nontranscriptional pathways [5]. Previous studies have shown that chronic treatment with ALDO could result in aortic endothelium dysfunction in normotensive and hypertensive rats due to the activation of COX-2, and could induce a vascular inflammatory phenotype in the rat myocardium and vascular smooth muscle cells through a mechanism causing an increase in COX-2 expression [17,18,19]. An in vitro study showed that ALDO increases the permeability of the human umbilical vein endothelial cell (HUVEC) monolayer for 60 min by the destabilization of intracellular junctions [20]. Another study showed that a high dose of intravenous administration of ALDO resulted in retinal edema in rats, which may indicate that ALDO increases the permeability of the blood–retina barrier [21]. However, the role of ALDO and MR in the skin vasculature is still not fully understood.

The present study evaluated the effect of ALDO on the permeability of skin microvessels in rats. In addition, the study analyzed the expression of the following: (1) MR in the skin tissue; (2) HSD11β2, which determines the selectivity of MR for ALDO; (3) vascular endothelial growth factor (VEGF), which is known as a proinflammatory factor responsible for increasing vascular permeability; (4) von Willebrand factor (vWF), which is an endothelial dysfunction marker and inflammatory mediator and can increase vascular permeability by the destabilization of tight junctions; and (5) zonula occludens 1 (ZO-1), a peripheral membrane protein responsible for vascular integrity [13,22,23,24].

## 2. Materials and Methods

### 2.1. Animals

All procedures involving animals were carried out in accordance with the institutional guidelines that are in compliance with the EU directive 2010/63/EU on the protection of animals used for scientific purposes and with the guidelines for the care and use of laboratory animals in biomedical research [25]. Approval for the study procedures was obtained from the Local Ethical Committee of Animal Testing at the Medical University of Bialystok (Approval Number 9/2016).

Wistar rats weighing 350–400 g were used in this study. The number of animals used in each experimental series is indicated in the figure legends. Because the age and weight of the animals may influence vascular permeability, the effect of these variables was minimized by using animals of similar size and age [26]. The animals were housed in a room with a 12 h dark/light cycle at a constant temperature and humidity. The animals were fed standard rat chow and provided tap water. They were deprived of food 24 h before the experiment but allowed free access to tap water. In order to minimize the effect of daily fluctuations of serum ALDO levels, all experiments using the rats were performed at the same time of day (9 AM) [27].

The rats were anesthetized by an intraperitoneal injection of pentobarbital sodium (45 mg/kg; Morbital, Biowet, Poland) and were euthanized immediately by pentobarbital overdose after skin removal, followed by exsanguination.

### 2.2. Experimental Protocol

In an acute experiment, eplerenone (EPL), a selective MR antagonist (100 mg/kg; Inspra; Pfizer, New York, NY, USA) or its solvent (CON; 5% gum arabic aqueous solution) was administered to rats in a single oral dose 30 min before the intradermal injection of ALDO (Sigma-Aldrich, Burlington, MA, USA) or its solvent (VEH, a mixture of saline and 0.14–14 ppm of ethanol). In a chronic experiment, to reflect clinical pharmacological treatment, EPL or CON was chronically administered to the rats for 10 consecutive days before ALDO/VEH injections. Indomethacin (IND; 8 mg/kg; Sigma-Aldrich, Burlington, MA, USA), a nonselective inhibitor of COX or CON, was administered in a single oral dose in a 5% gum arabic solution 60 min prior to the ALDO/VEH injections.

The back of each of the rats was shaved and divided symmetrically with respect to the spine into four areas with even surfaces (1.5 × 1.5 cm). ALDO (1, 10, or 100 pmol), histamine (HIST; 9 μmol saline solution, Sigma-Aldrich, Burlington, MA, USA), or VEH was randomly injected intradermally into the center of each area in a 0.1 mL volume (each animal received both ALDO and VEH; *n* for each group is the number of skin samples, not the number of animals). From each rat, four skin samples, with different injected agents, were obtained. Thus, each experimental condition was tested in at least eight animals (Figure 1A). For further investigation, the skin and blood samples were collected 30 min after the intradermal injections of ALDO/VEH (Figure 1B).

### 2.3. Vascular Permeability Measurement

The well-established Miles assay method was used to measure vascular permeability [28]. The principle of this method is that agents affecting vascular permeability enhance the diffusion of the Evans blue dye into the tissue injected with a studied substance. Evans blue dye (30 mg/kg; Sigma-Aldrich, Burlington, MA, USA) was administered into the femoral vein 15 min after the intradermal injection of ALDO/VEH (Figure 1). After 15 min of dye injection, skin samples were taken, weighed, eluted with formamide (4 mL), and then incubated at 45 °C for 72 h. Next, the concentration of the dye in the supernatants was determined colorimetrically (λ = 620 nm). The results were presented as the Evans blue concentration per gram of the skin tissue. To validate the method followed, HIST, which is a known factor in increasing vascular permeability in this model, was injected as a positive control in the same way as ALDO/VEH [29,30].

### 2.4. Histological and Immunohistochemical (IHC) Analysis

Skin biopsy samples were fixed in 4% phosphate-buffered formaldehyde and embedded in paraffin. For a general histological evaluation, 4 μm thick sections were cut and stained with hematoxylin and eosin (H + E). The histological evaluation was conducted by an experienced histologist in a blinded manner, using an OLYMPUS analysis set (BX50 Microscope, DP20 Camera, Cell^D^ Imaging Software; OLYMPUS, Tokyo, Japan). In order to evaluate the changes in the thickness of specific skin layers, the thickness of the epidermis, dermis, and hypodermis layers was measured and expressed as a percentage of total skin thickness.

The paraffin-embedded sections were used for IHC staining. IHC analysis was carried out to study the presence of vWF, VEGF, MR, HSD11β2, and ZO-1, factors that could be involved in the regulation of vascular permeability. All procedures were performed using commercially available kits according to the manufacturer’s instructions. Specific antibodies for vW (kit No. M0616; DakoCytomation, Glostrup, Denmark), VEGF (kit No. M7273; DakoCytomation, Glostrup, Denmark), MR (kit No. ab2774; Abcam, Cambridge, UK), HSD11β2 (kit No. ab203132; Abcam, Cambridge, UK), and ZO-1 (kit No. 214228; Abcam, Cambridge, UK) were used for the analysis.

If the reaction was positive, brown-colored antigen–antibody complexes appeared at the position of the target molecule. The intensity of the positive reaction staining signal was graded on a scale from (−) to (+++), with (−) representing no detectable signal; (+), a slight signal; (++), a moderate signal; and (+++), a strong signal. The positive-stained cells were examined under high-powered fields with the same OLYMPUS analysis set.

### 2.5. mRNA Extraction and Real-Time Quantitative Polymerase Chain Reaction (rt-qPCR)

The mRNA expression of vWF, VEGF, MR, HSD11β2, and ZO-1 was analyzed by rt-qPCR. For this, skin samples were homogenized, and total RNA was extracted using the Trizol-based protocol (1 mL per 100 mg of skin tissue; Thermo Fisher Scientific, Waltham, MA, USA), with additional washing with 75% ethanol. The total RNA quantity was determined spectrophotometrically, and 1 μg of RNA was DNase-treated (Dnase I Amplification Grade; Sigma-Aldrich, Burlington, MA, USA) prior to the removal of double- and single-stranded DNA. Reverse transcription was carried out using the *SensiFAST*™ *cDNA Synthesis Kit* (Bioline, London, UK). rt-qPCRs were performed with *SsoAdvanced Universal SYBR Green Supermix* (Bio-Rad, Hercules, CA, USA) using a Stratagene Mx3005P thermocycler (Agilent Technologies, Santa Clara, CA, USA) under the following conditions: 10 min at 95 °C, 40 cycles of 30 s at 95 °C, 60 s at 60 °C, and 30 s at 72 °C. The expression levels of all genes of interest were normalized to the housekeeping gene β-actin (*Actb*). The relative mRNA expression was calculated using the qBase MS Excel VBA applet [31]. The 5′—3′ sequence of the primers used was as follows: VEGF (F: CATCAGCCAGGGAGTCTGTG; R: GAGGGAGTGAAGGAGCAACC); vWF (F: GCTCAGGGACATGGCTTAGG; R: CCATACAAACAGGGGCCGTA); MR (F: CCATGCAGGCAACATTACCG; R: GTAAGAAAGGCCCCACCCTC); HSD11β2 (F: CCAGCCACATGGAAGCTGTA; R: CAAACACTATCTCTCCCATTCTAGG); ZO-1 (F: GGTAGTGCAAAGAGATGAGC; R: GGCATTAGCAGAATGGATAC); and Actb (F: GCAGGAGTACGATGAGTCCG; R: ACGCAGCTCAGTAACAGTCC).

### 2.6. Blood Morphology

Blood morphology was analyzed using the hematological analyzer ScilVet ABC Plus+ (Horiba ABX, Montpellier, France). Blood cells were counted by following the volumetric impedance method. Direct measurements of white blood cells, red blood cells, hemoglobin, and platelets were performed, followed by the automatic calculation of hematocrit.

### 2.7. Serum and Skin ALDO Concentrations

The levels of ALDO in the serum and the skin were determined using ELISA Kit No. 501090 (Cayman Chemicals, Ann Arbor, MI, USA) according to the manufacturer’s instructions.

The skin ALDO concentration was measured based on the method described by Reynoso-Palomar et al., with some modifications [32]. Briefly, skin fragments were homogenized in phosphate-buffered saline (1 mL per 100 mg of the skin tissue). The homogenate cell membranes were broken by two freeze–thaw cycles. Then, the chloroform extraction protocol was performed in accordance with the kit manufacturer’s protocol. After the evaporation of chloroform, the obtained extract was dissolved in ELISA buffer, and the ALDO concentration in each sample was measured according to the kit manufacturer’s protocol.

### 2.8. Statistical Analysis

Data were presented as mean ± SEM. The Shapiro–Wilk normality test was performed to test the distribution of the data. A one-way analysis of variance with Bonferroni multiple comparison post hoc correction was used to identify statistical differences between the groups. *p* values < 0.05 were considered statistically significant. Statistical analysis was performed using GraphPad Prism v. 8.0.1 software (GraphPad Software, San Diego, CA, USA).

## 3. Results

### 3.1. ALDO Increased Vascular Permeability

In the first set of experiments, the effect of acute ALDO injection on vascular permeability was evaluated. It was observed that only the highest dose of ALDO caused a significant increase in vascular permeability (a 41% increase; *p* < 0.05). The effects of 1 and 10 pmol ALDO doses were not statistically significant; however, these doses were associated with a tendency to increase vascular permeability. For further investigation, 100 pmol of ALDO was used. HIST (positive control) increased vascular permeability by 106% (*p* < 0.001), which confirmed the validity of the experiment (Figure 2).

#### 3.1.1. Acute EPL and IND Administration Diminished the ALDO-Increased Vascular Permeability

The next set of experiments determined whether acute EPL administration diminishes the increase in the vascular permeability induced by ALDO. It was observed that ALDO enhanced vascular permeability by 33% (*p* < 0.01), whereas acute EPL administration 30 min prior to intradermal ALDO injection prevented this effect (*p* < 0.05) (Figure 3).

To verify whether the ALDO-elevated vascular permeability is a COX-dependent phenomenon, IND was administered. The effect of ALDO in the IND-treated group was found to be 33% lower than in the CON + ALDO group (*p* < 0.001) (Figure 3); however, acute IND administration by itself resulted in a 24% reduction in basal skin vascular permeability (*p* < 0.05).

#### 3.1.2. Chronic EPL Administration Reduced Basal Vascular Permeability

To determine whether chronic EPL administration changes vascular permeability, EPL was administered for 10 days in the next set of experiments. Chronic EPL administration was performed to reflect clinical pharmacological treatment. In ALDO-injected skin, a 30% increase in vascular permeability was observed (*p* < 0.05). The ALDO-induced increase in vascular permeability was arrested after 10 days of EPL administration (*p* < 0.001). Moreover, a 31% reduction in basal vascular permeability was observed after chronic EPL administration (*p* < 0.05) (Figure 4).

### 3.2. Histological and Morphometric Analysis

To evaluate whether ALDO-induced vascular leakage is accompanied by local edema, or whether EPL administration changes the thickness of particular skin layers, a histological and morphometric analysis was carried out. In H + E staining images, no changes were observed in the histological appearance of the studied groups (Appendix A). In addition, no changes in skin thickness were observed in the ALDO- and EPL-treated groups (Appendix A).

### 3.3. Blood Morphology

Chronic EPL administration did not affect blood morphology (Appendix A).

### 3.4. IHC Staining for vWF, VEGF, MR, HSD11β2, and ZO-1

To evaluate whether acute ALDO injection or EPL administration affects the staining strength of selected factors in skin blood vessels, IHC analysis was carried out. We did not observe staining for HSD11β2 in skin blood vessels in any of the studied groups.

#### 3.4.1. Acute ALDO Administration Activated Endothelial vWF Exocytosis

In the next set of experiments, a single EPL or CON dose was administered 30 min before the injection of ALDO/VEH. No staining (−) from vWF was observed in the ALDO-injected skin (CON + ALDO group). Acute administration of EPL did not change the vWF staining strength (++); however, it blocked the ALDO effect and restored moderate (++) staining in the EPL + ALDO group (Figure 5).

#### 3.4.2. Chronic EPL Administration Led to vWF Accumulation in Blood Vessels and VEGF Staining Reduction

In another set of experiments, EPL was chronically administered for 10 days before ALDO injection. Moderate staining (++) from vWF was observed in the CON + VEH group, whereas a lack of staining (−) was observed in the CON + ALDO group. Chronic EPL administration resulted in strong staining (+++) in both the VEH- and ALDO-injected groups. No reduction in staining strength from vWF after ALDO injection was observed in the group chronically treated with EPL (Figure 6). Moreover, in both the VEH- and ALDO-injected groups, chronic EPL administration reduced the VEGF staining strength in the skin vasculature (from (++) to (+)). ALDO injection had no effect on the VEGF staining strength (Figure 7). No detectable differences in MR, HSD11β2, and ZO-1 staining were observed between the studied groups.

#### 3.5. mRNA Level of vWF, VEGF, MR, HSD11β2, and ZO-1

The analysis of mRNA expression showed the presence of vWF, VEGF, MR, HSD11β2, and ZO-1 in the skin samples. No differences in gene expression were observed between the groups (Table 1).

### 3.6. EPL Increased Serum and Skin ALDO Concentrations

Chronic EPL administration resulted in a 108% increase in plasma (*p* < 0.001) and a 45% increase in the skin ALDO concentration (*p* < 0.05) (Figure 8).

## 4. Discussion

### 4.1. Used Model

A well-established vascular permeability measurement method, based on the binding of Evans blue dye to plasma proteins, was used in this study, which allowed the colorimetric measurement of protein extravasation after ALDO intradermal injection [28,29]. To block the effect of ALDO, EPL was administered 30 min before its injection. Pharmacokinetic studies have shown that 30 min is a sufficient duration for ALDO to reach its maximal plasma concentration after oral administration in rats [33]. Similarly, our previous studies revealed that oral EPL administration (100 mg/kg) 30 min prior to ALDO injection is sufficient to block its action in blood vessels [34,35]. Moreover, in a rat model, EPL (a single 20 mg/kg oral dose) accumulated in the skin tissue, and after 1 h, reached an almost thrice higher concentration than in blood; thus, its effect can be manifested in skin tissue microcirculation [36].

### 4.2. Acute ALDO Effects on Vascular Permeability

This was the first in vivo model to provide direct evidence that acute ALDO administration increases vascular permeability in rat skin, whereas EPL administration completely prevents the action of ALDO. Therefore, the acute effect of ALDO on vascular permeability may result from the activation of MR. A similar effect was observed under in vitro conditions, where ALDO increased the permeability of dextran (70 kDa) into the HUVEC monolayer within 60 min by the redistribution of intracellular junction structures. The MR blockade attenuated the ALDO-related effects [20]. These ALDO effects were also confirmed ex vivo in freshly isolated umbilical arteries [20]. The ALDO-increased paracellular permeability was accompanied by a tight junction rearrangement process, manifested by the formation of F-actin stress fibers and the disruption of zonula occludens (ZO)-1 junction strands. ZO-1 plays a key role in the dynamic regulation of cell–cell integrity by anchoring tight junction proteins to the cytoskeleton and providing tension to the adherens junctions [13]. In our study, no changes were observed in mRNA expression and IHC staining for ZO-1 in the in vivo evaluation of the acute effects of ALDO on the endothelium of skin vessels. Although no changes in the expression of ZO-1 were observed in our studies, the involvement of this protein in the regulation of MR-dependent vascular permeability cannot be clearly ruled out, as the microstructural distribution of ZO-1 within endothelial cells was not evaluated in our study. In contrast to the above results, the preincubation of HUVECs with ALDO for 3 days did not change the transendothelial ion and dextran permeability, while both apical cell surface and apical cell stiffness were increased. The MR antagonist prevented these responses [37]. The exact mechanism of the interaction of ALDO with endothelial cells still remains to be investigated.

Our study confirmed the involvement of prostaglandins in vascular permeability during inflammation [38]. Similarly to the previous observation [30], we observed that IND (COX-1/COX-2 inhibitor) decreases the initial vascular permeability in rat skin. It was also shown that the effect of ALDO in the IND-treated group was lower than in the CON + ALDO group. This suggests that vascular permeability may also be constitutively regulated by COX, and the effects of ALDO are limited while COX is deactivated.

VEGF increases microvascular endothelial cell permeability via nitric oxide and prostacyclin, as nitric oxide synthase and COX inhibitors attenuate its vascular effect [23]. Our study was the first to evaluate, in vivo, the VEGF-mediated effect of ALDO on vascular permeability. In our rat model, no changes were found in the IHC signal strength from VEGF in the acute ALDO-injected skin. Interestingly, the acute effect of ALDO on VEGF production was observed in vitro in human neutrophils 30 min after the addition of ALDO, and the effect peaked after around 6 h. This effect was only partially diminished by RNA and protein synthesis inhibitors, as well as by MR antagonists, confirming the involvement of the rapid pathway in ALDO-increased VEGF production [39]. It was also reported that 48 h of incubation with ALDO increased the endothelial expression of VEGF in mouse cultured cortical collecting duct epithelial cells in an MR-dependent mechanism [40]. In contrast, ALDO (10 to 1000 nM) given for 4 days reduced the mRNA levels of VEGF receptor 2, without having any effect on the production of VEGF or mRNA levels of VEGF in bone marrow-derived endothelial progenitor cells [41]. It seems that the differences in the effect of ALDO on VEGF expression may be related to the models used (in vitro/in vivo, tissue and cell type, time of incubation, and ALDO concentration). Nevertheless, the role of VEGF in the mechanism of ALDO-increased vessel permeability still remains to be examined.

vWF is stored in Weibel–Palade bodies (WPB) in endothelial cells, where it remains inactive, while in endothelial dysfunction, it is excreted from WPB [42]. In the present study, ALDO injection diminished the IHC signal of vWF within 30 min, which suggests the exocytosis of WPB from endothelial cells. A similar mechanism was observed in an earlier study in a human aortic endothelial cell (HAEC) culture, where ALDO activated WPB exocytosis, leading to the release of vWF and P-selectin within 10 min. This effect was not abolished by actinomycin D (a DNA transcription inhibitor), which may suggest the nongenomic mechanism of ALDO action [43]. Corresponding data from our previous study showed that the vWF concentration in HUVEC culture supernatants was increased 10 min after ALDO exposure. The additional preincubation of HUVECs with ALDO for 120 min further increased the secretion of vWF [35]. In this study, acute administration of EPL diminished the exocytosis of vWF, which indicates an MR-dependent mechanism. Similarly, in HAECs, 1 h of incubation with spironolactone (a nonspecific MR antagonist) inhibited ALDO-induced vWF exocytosis [43]. Furthermore, the level of vWF was shown to be increased in patients with PA [44]. The results of the present study seem to suggest the interplay between high ALDO levels and vWF exocytosis.

Our study, for the first time, showed the expression of MR in the rat skin vasculature, although no changes were observed in the mRNA expression and IHC staining for MR after the administration of ALDO or EPL. MR is not selective to ALDO. MR and GR are members of the same nuclear receptor subfamily, NR3C, which can be activated by both mineralocorticoids and glucocorticoids [45,46]. The physiological plasma concentration of the glucocorticoid cortisol (CORT) is up to 1000-fold higher than that of ALDO. Thus, to prevent permanent MR occupancy by CORT, coexpression of HSD11β is required. 11βHSD can metabolize CORT to its inactive form—corticosterone (CTC). A previous study showed that type 1 isoform HSD11β1 is responsible for CORT–CTC transformation in both directions [22]. However, in skin tissue, HSD11β acts predominantly as a reductase, deactivating CORT [47,48]. Similarly, type 2 isoform (HSD11β2) only catalyzes CORT deactivation and is expressed exclusively in the highly mineralocorticoid-selective tissues [49]. The low mRNA expression of HSD11β2 has been confirmed in HAECs [50]. In this study, the expression of HSD11β2 was not observed in the skin vasculature. However, MR selectivity to ALDO may also be conferred by type 1 isoenzyme, which is not the subject of the present study. The lack of changes in the selected gene expression may be partially explained by acute 30 min ALDO effects, and thus the rapid mechanism of hormone action [14].

### 4.3. Chronic EPL Effect on Vascular Permeability

In the present study, chronic EPL administration showed a strong effect on endothelium per se, which is manifested by a reduction in permeability. Previously, the effect of EPL in the reduction in vascular permeability was observed in rat choroidal vasculature, as well as in patients with choroid retinopathy, where EPL treatment reduced macular edema and the amount of subretinal fluid [51].

Our findings showed that EPL administration increases vWF endothelial accumulation. Despite the increased vWF signal in blood vessels, no differences were observed in vWF gene expression in skin homogenates. This increased signal from vWF may be due to the inhibition of constitutive WPB exocytosis. It was shown that exogenous ALDO induces the endothelial exocytosis of WPB and the release of vWF in HAECs, while spironolactone antagonizes the ALDO effect [43].

We observed that chronic EPL administration reduced the VEGF staining strength in the skin vasculature. A decrease in VEGF expression after chronic EPL administration was also observed in rat glomeruli [52]. However, the data on the MR–VEGF interaction are contrasting. It was found in a mouse model that MR activation inhibited VEGF-induced gene expression, reducing neoangiogenesis [53]. Clinical data suggest the opposite trend, as the MR blockade limited choroidal neovascularization, with an effectiveness comparable to that of anti-VEGF therapy [51]. In conclusion, the reduction observed in vascular permeability after chronic EPL treatment may possibly be due to the reduction in constitutive VEGF expression in blood vessels. Another theory may point to the pleiotropic effects of EPL. EPL is known for its anti-inflammatory potential, manifested mainly through the reduction in inflammatory cytokines [54,55], and it is assumed that those effects go far beyond the mechanism of the classic MR blockage [56].

However, there are limitations to our study. First, the doses of ALDO used in this study were supraphysiological, since the physiological plasma ALDO concentration goes from 55 to 250 pmol/L [57]. However, even higher dose ranges have been used in many in vitro as well as in vivo studies investigating the mechanism of ALDO action [20,35,43,51]. ALDO levels can be extremely increased in many human pathological states, such as hypertension, primary hyperaldosteronism, heart failure, and diabetes; moreover, increased levels of ALDO were observed in patients and animals undergoing intra-abdominal surgery during the procedure, as well as in the postoperative period [58,59,60]. Second, the IHC staining was performed to evaluate the expression of commonly used vascular permeability markers. IHC staining is a well-established, widely accepted method in both clinical and experimental parts of medical science. However, the lack of standardization, especially on the postanalytical stage (the interpreting and reporting of results), makes the comparison of the results of different studies quite difficult. Moreover, the staining controls for IHC should be included to support qualitative results, but we could not provide them in this paper, as this would require additional animal experiments. Finally, the detection of secreted VEGF proteins by relying on staining patterns from a single antibody is considered problematic. For VEGF staining, we used antibodies that are commonly used in studies on rats and mice [61,62,63,64,65]. However, the IHC results should be confirmed with some other method (e.g., immunoblotting and real-time PCR).

## 5. Conclusions

The results of this study confirm that ALDO regulates skin vascular physiology via an MR-dependent mechanism. ALDO injection resulted in a significant increase in vascular permeability and enhanced the endothelial exocytosis of vWF. This effect of ALDO was blocked by EPL administration. Chronic EPL administration led to vWF accumulation and a reduction in the IHC staining for VEGF. However, ALDO or EPL had no effect on the mRNA expression of the studied genes or skin structure. This study showed, for the first time, the presence of MR in the rat skin vasculature.

From the clinical point of view, the ALDO-increased skin vascular permeability, accompanied by endothelial dysfunction, suggests the role of ALDO in skin vascular disorders observed in diseases characterized by increased ALDO levels, such as diabetes and PA. Increased vascular permeability has been observed in diabetic dermatopathy, which is the most common dermal complication of diabetes. Our preliminary results showed that chronic EPL administration reduced skin vascular permeability in STZ-diabetic rats. Thus, the prevention of or reduction in ALDO-induced vascular changes with MR blockade may account for the clinical benefits.

## Figures and Tables

**Figure 1 cells-11-02707-f001:**
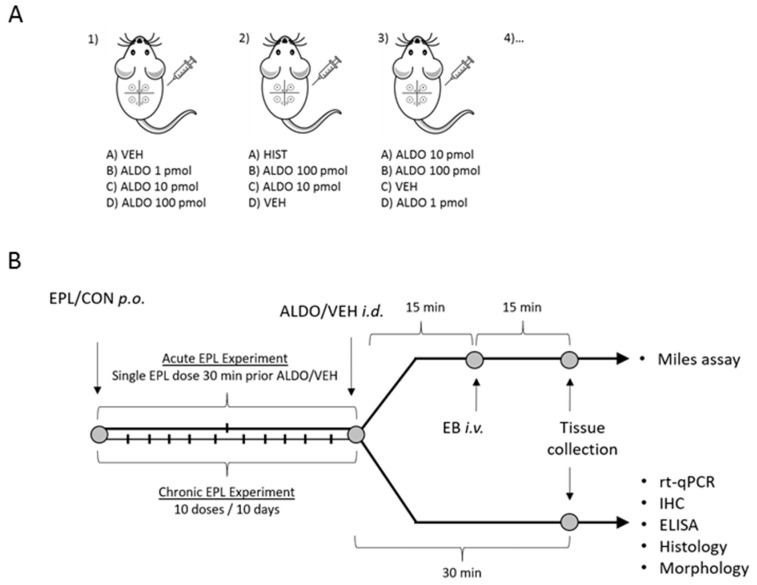
Experimental protocol; (**A**) intradermal injection of tested agents; (**B**) experimental timeline. ALDO—aldosterone; CON—eplerenone solvent; EB—Evans blue; ELISA—enzyme-linked immunosorbent assay; EPL—eplerenone; HIST—histamine; i.d.—intradermal; IHC—immunohistochemistry; i.v.—intravenous; p.o.—per os; rt-qPCR—real-time quantitative polymerase chain reaction; VEH—ALDO solvent.

**Figure 2 cells-11-02707-f002:**
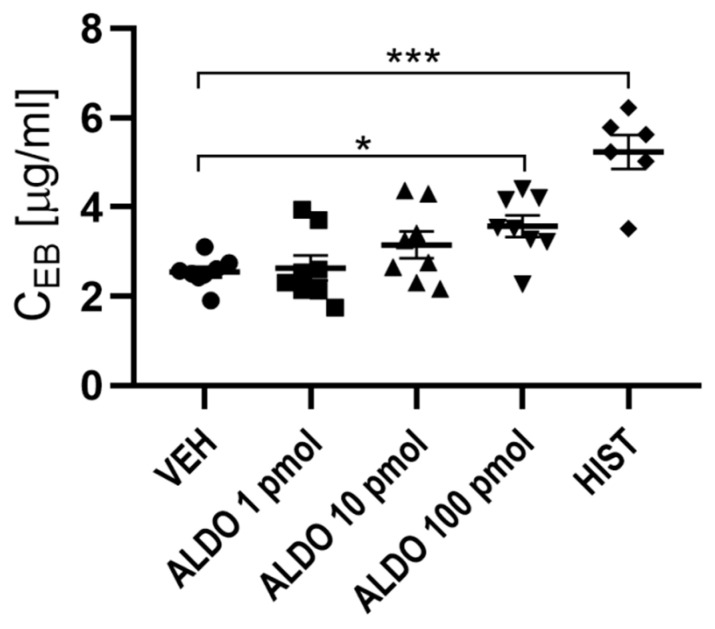
Dose-dependent effect of ALDO on vascular permeability. *n* = 6–8; *n* for each group is the number of skin samples, not the number of animals; 10 rats were used in this experiment; data are presented as mean ± SEM and individual values; HIST was used as a positive control. ALDO—aldosterone; C_EB_—Evans blue concentration per gram of skin tissue; HIST—histamine; VEH—ALDO solvent; * *p* < 0.05; *** *p* < 0.001.

**Figure 3 cells-11-02707-f003:**
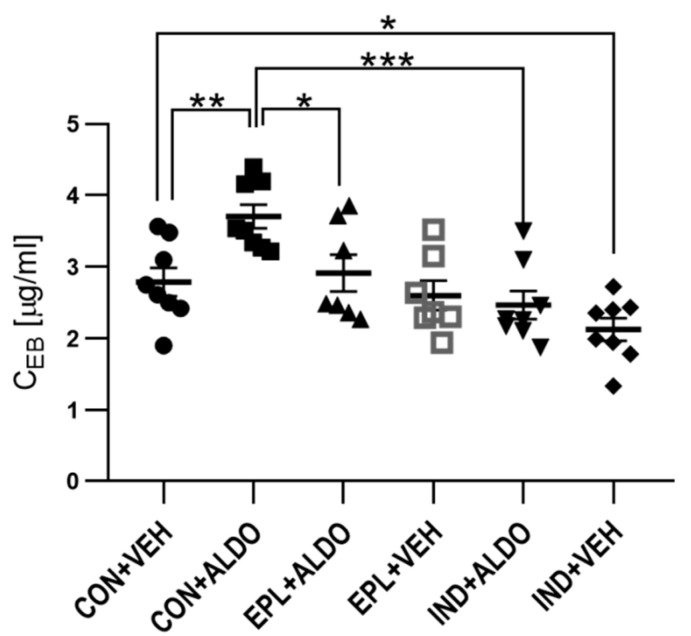
Changes in vascular permeability after acute EPL or IND administration. *n* = 7–8; *n* for each group is the number of skin samples, not the number of animals; 12 rats were used in this experiment; data are presented as mean ± SEM and individual values; ALDO—aldosterone; C_EB_—Evans blue concentration per gram of skin tissue; CON—5% gum arabic solution; EPL—eplerenone; IND—indomethacin; VEH—ALDO solvent; * *p* < 0.05; ** *p* < 0.01; *** *p* < 0.001.

**Figure 4 cells-11-02707-f004:**
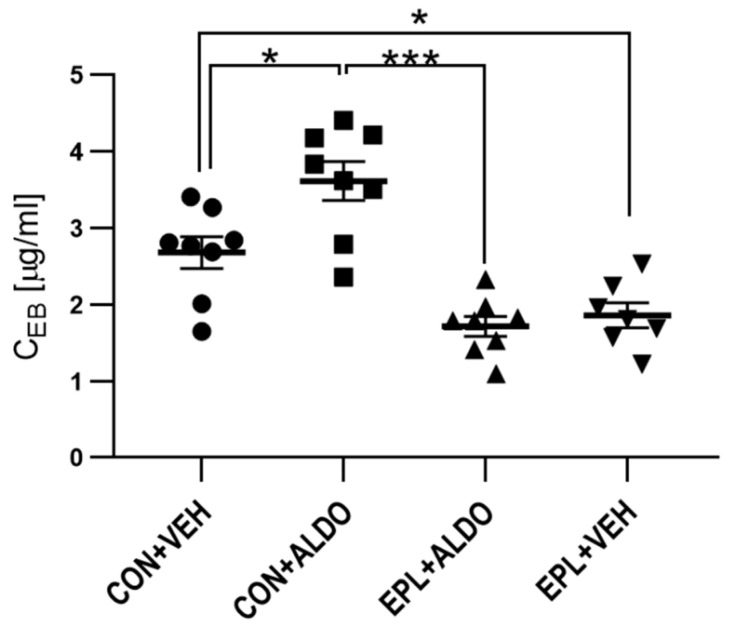
Changes in vascular permeability after chronic EPL administration. *n* = 7–8; *n* for each group is the number of skin samples, not the number of animals; 8 rats were used in this experiment; data are presented as mean ± SEM and individual values; ALDO—aldosterone; C_EB_—Evans blue concentration per gram of skin tissue; CON—EPL solvent; EPL—eplerenone; VEH—ALDO solvent; * *p* < 0.05; *** *p* < 0.001.

**Figure 5 cells-11-02707-f005:**
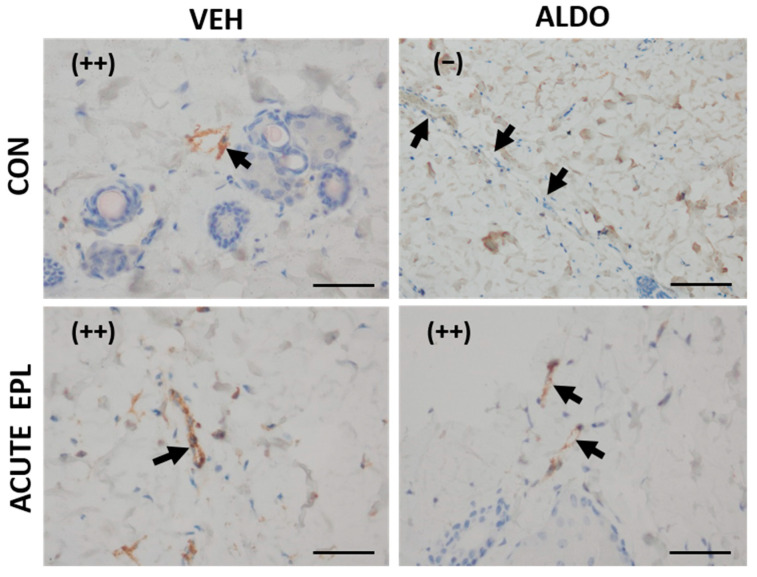
ALDO induced endothelial exocytosis of vWF in skin blood vessels. *n* = 8; *n* for each group is the number of skin samples, not the number of animals; 16 rats were used in this experiment; immunohistochemical staining for von Willebrand factor showed a lack of staining (−) in ALDO-injected skin. Acute EPL administration restored the vWF staining strength (++) in ALDO-injected skin. ALDO—aldosterone; CON—5% gum arabic solution; EPL—eplerenone; VEH—ALDO solvent. (−)—lack of staining; (++)—moderate staining. Arrows point to blood vessels. Scale bars: 50 μm.

**Figure 6 cells-11-02707-f006:**
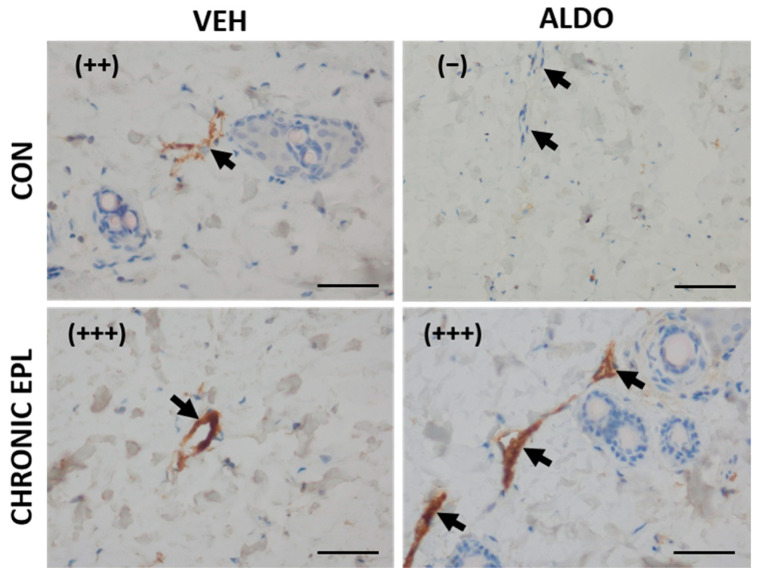
Chronic EPL administration led to vWF accumulation in the blood vessels. *n* = 8; *n* for each group is the number of skin samples, not the number of animals; 32 rats were used in this experiment; immunohistochemical staining for von Willebrand factor showed a lack of staining (−) in ALDO-injected skin. Chronic EPL administration increased staining strength (+++) and prevented the ALDO effect. ALDO—aldosterone; CON—5% gum arabic solution; EPL—eplerenone; VEH—ALDO solvent. (−)—lack of staining; (++)—moderate staining; (+++)—strong staining. Arrows point to blood vessels. Scale bars: 50 μm.

**Figure 7 cells-11-02707-f007:**
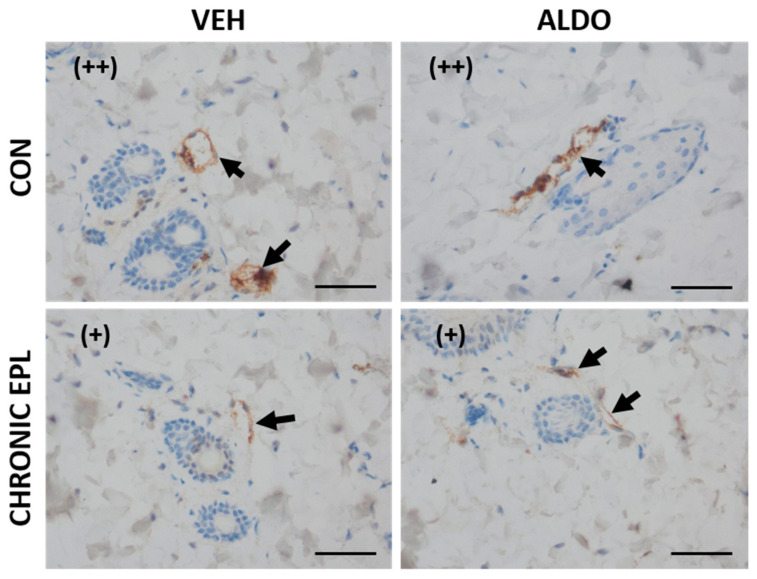
Chronic EPL administration led to VEGF staining reduction in the skin vasculature. *n* = 8; *n* for each group is the number of skin samples, not the number of animals; 32 rats were used in this experiment; immunohistochemical staining for VEGF was not changed in ALDO-injected skin (++). Chronic EPL administration decreased signal strength (+). ALDO—aldosterone; CON—5% gum arabic solution; EPL—eplerenone; VEH—ALDO solvent. (+)—slight staining; (++)—moderate staining. Arrows point to blood vessels. Scale bars: 50 μm.

**Figure 8 cells-11-02707-f008:**
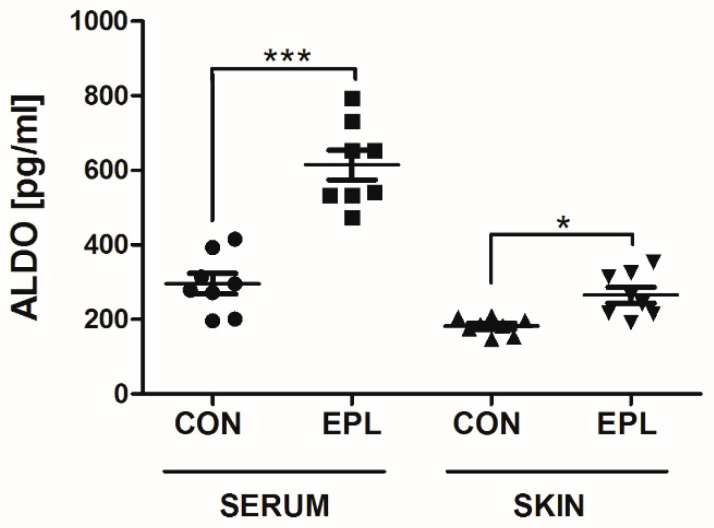
Changes in the serum and skin ALDO concentrations after chronic EPL administration. *n* = 8; *n* for each group is the number of skin samples, not the number of animals; 32 rats were used in this experiment; data are presented as mean ± SEM and individual values; ALDO—aldosterone; CON—5% gum arabic solution; EPL—eplerenone; * *p* < 0.05; *** *p* < 0.001.

**Table 1 cells-11-02707-t001:** Relative expression of the studied genes normalized to β-actin expression.

Group		vWF/Actb	VEGF/Actb	MR/Actb	HSD11β2/Actb	ZO-1/Actb
CON	VEH	18.04 ± 0.69	19.47 ± 1.13	22.00 ± 3.93	18.91 ± 1.07	19.09 ± 1.07
	ALDO	19.16 ± 0.95	20.60 ± 0.95	17.60 ± 1.34	20.30 ± 1.13	21.17 ± 1.11
Acute EPL	VEH	17.75 ± 1.33	21.22 ± 0.84	22.47 ± 5.34	20.29 ± 0.95	19.18 ± 1.45
	ALDO	13.92 ± 2.97	19.64 ± 0.76	22.48 ± 4.83	18.94 ± 0.51	18.40 ± 1.16
CON	VEH	20.57 ± 0.12	14.81 ± 1.44	18.03 ± 0.67	19.35 ± 0.16	17.96 ± 0.52
	ALDO	20.67 ± 0.24	17.07 ± 0.51	19.25 ± 0.61	19.51 ± 0.07	19.27 ± 0.46
Chronic EPL	VEH	20.42 ± 0.13	17.12 ± 1.25	17.99 ± 0.56	19.41 ± 0.12	18.56 ± 0.39
	ALDO	20.10 ± 0.27	16.90 ± 0.70	18.39 ± 0.61	19.46 ± 0.15	17.81 ± 0.40

*n* = 6–8; *n* for each group is the number of skin samples, not the number of animals; 32 rats were used in this experiment; data are presented as mean ± SEM; Actb—β-actin; ALDO—aldosterone; CON—5% gum arabic solution; EPL—eplerenone; HSD11β2—11β-hydroxysteroid dehydrogenase type 2; MR—mineralocorticoid receptor; VEGF—vascular endothelial growth factor; VEH—ALDO solvent; vWF—von Willebrand factor; ZO-1—zonula occludens-1.

## Data Availability

The raw data supporting the findings of this manuscript will be provided by the authors at any time to the reviewers, and thereafter, to any researcher after publication.

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
