# Peer review of "Aldosterone Increases Vascular Permeability in Rat Skin"

_cells, 2022, doi:10.3390/cells11172707_

Round 1
Reviewer 1 Report
In the present manuscript, the authors evaluated the effect of aldosterone on skin vascular permeability. Towards this goal they intradermally injected aldosterone into rats and then measure vascular permeability. The authors observed a 30% increase in vascular permeability after aldosterone administration; this effect was reverted after eplerenone administration.
The manuscript is well-organized. The topic is sound. The methodology is clearly reported. Conclusions are supported by the results. I have only minor comments:
- According to the experimental design, the effect of aldosterone was evaluated after 30 min; why the authors chose this time point? They also have data on a long-term exposition? If not, please discuss.
- I did not find eplerenone dose; please specify.
- According to IHC, aldosterone induced endothelial exocytosis of vWF, while chronic EPL administration led to vWF accumulation in the blood vessels and reduced VEGF staining reduction in the skin vasculature (Figures 6-8); may the authors report quantitative data and level of significance on these effects.
- The use of supraphysiological dose of aldosterone should be acknowledge also as limitation of the present study.
Author Response
Comments and Suggestions for Authors
In the present manuscript, the authors evaluated the effect of aldosterone on skin vascular permeability. Towards this goal they intradermally injected aldosterone into rats and then measure vascular permeability. The authors observed a 30% increase in vascular permeability after aldosterone administration; this effect was reverted after eplerenone administration.
The manuscript is well-organized. The topic is sound. The methodology is clearly reported. Conclusions are supported by the results. I have only minor comments:
- According to the experimental design, the effect of aldosterone was evaluated after 30 min; why the authors chose this time point? They also have data on a long-term exposition? If not, please discuss.
- The effect of aldosterone was evaluated after 30 min, although the Evans blue dye for vascular permeability measurement was administered 15 min after aldosterone injection. In our studies we are focusing mainly on acute, rapid actions of hormone in the vasculature, known as non-genomic effects, that occur within minutes (5 min - 30 min) and do not require transcription or translation [Gromotowicz-Poplawska et al., Vascul Pharmacol 2019; Gromotowicz-Poplawska et al., J Am Heart Assoc 2021]. Recently, considerable attention has been focused on rapid effects of aldosterone on the vascular wall. There are numerous studies describing aldosterone effects on the vascular activity or endothelium function [Toda et al., Br J Pharmacol 2013; Ruhs et al.,
J Endocrinol 2017], although there are no data regarding rapid effects of aldosterone on skin vasculature. Thus, we have started with the evaluation of the acute aldosterone effects in rat skin, and we are going to perform studies with long-term exposition to high aldosterone level as well.
- I did not find eplerenone dose; please specify.
- Eplerenone was used at a dose of 100 mg/kg , it was mentioned in the Materials and Methods section (2.2 Experimental protocol, page 3, line 107). This dose was used also in our previous studies, what was discussed this in the first paragraph of Discussion (page 12; lines 349-351).
- According to IHC, aldosterone induced endothelial exocytosis of vWF, while chronic EPL administration led to vWF accumulation in the blood vessels and reduced VEGF staining reduction in the skin vasculature (Figures 6-8); may the authors report quantitative data and level of significance on these effects.
- IHC is a well-established, widely accepted method in both clinical and experimental parts of medical science. It allows receiving valuable information about any process in any tissue. Each year the amount of data, received by IHC, grows in geometric progression. But the lack of standardization, especially on the post-analytical stage (interpreting and reporting of results), makes the comparison of the results of different studies quite difficult. For IHC data a qualitative interpretation is commonly used among scientists as a “gold standard” [Fedchenko et al., Diagn Pathol 2014]. In addition to the description of the evaluated parameters scientists use qualitative scoring systems to interpret received data, usually the force of IHC staining in different investigated areas. Score ranks usually lie in a range from “negative” (mostly marked as “-”) to “positive”, which may be signed with different amount of “+” depending on how many other categories lay between these border parameters. Most common spectrum of categories, describing different force of IHC expression in investigated groups, include: “negative”(−), “slight”(+), “moderate”(++), “strong”(+++) and their variations. The histological evaluation was conducted by an experienced histologist in a blinded manner. This was of IHC data interpretation and presentation is widely accepted. The quantification of IHC data and statistics is very difficult, so usually other methods are used to confirm the IHC results. The limitation of the IHC staining we described at the end of the revised manuscript.
- The use of supraphysiological dose of aldosterone should be acknowledge also as limitation of the present study.
- As recommended, in the revised manuscript the use of supraphysiological dose of aldosterone was described as a limitation of the study (Discussion, p. 14; lines: 462-469).
References:
- Gromotowicz-Poplawska A, Marcinczyk N, Misztal T, Golaszewska A, Aleksiejczuk M, Rusak T, Chabielska E. Rapid effects of aldosterone on platelets, coagulation, and fibrinolysis lead to experimental thrombosis augmentation. Vascul Pharmacol. 2019 Nov-Dec;122-123:106598. doi: 10.1016/j.vph.2019.106598.
- Gromotowicz-Poplawska A, Flaumenhaft R, Gholami SK, Merrill-Skoloff G, Chabielska E, Williams GH, Romero JR. Enhanced Thrombotic Responses Are Associated With Striatin Deficiency and Aldosterone. J Am Heart Assoc. 2021 Nov 16;10(22):e022975. doi: 10.1161/JAHA.121.022975.
- Toda N, Nakanishi S, Tanabe S. Aldosterone affects blood flow and vascular tone regulated by endothelium-derived NO: therapeutic implications. Br J Pharmacol. 2013 Feb;168(3):519-33. doi: 10.1111/j.1476-5381.2012.02194.x.
- Ruhs S, Nolze A, Hübschmann R, Grossmann C. 30 YEARS OF THE MINERALOCORTICOID RECEPTOR: Nongenomic effects via the mineralocorticoid receptor. J Endocrinol. 2017 Jul;234(1):T107-T124. doi: 10.1530/JOE-16-0659.
- Fedchenko N, Reifenrath J. Different approaches for interpretation and reporting of immunohistochemistry analysis results in the bone tissue - a review. Diagn Pathol. 2014 Nov 29;9:221. doi: 10.1186/s13000-014-0221-9.
Reviewer 2 Report
The authors investigate the effect of aldosterone in the rat skin. At relatively high concentrations aldosterone appeared to induce permeability, which was not associated with major overall alterations in protein expression. Involvement of MR and COX pathways was tested using pharmacological inhibitors.
The study results are potentially interesting. They do not agree much with published data (from other mainly in vitro systems). However, value of the presented data is vastly diminished by insufficient n numbers and the use fo antibodies that have not been validated for IHC.
Specifically
Results:
All figures: I applaude the authors for being very precise about the number of animals used. However, what they claim to be n numbers are technical replicates. By my reckoning each exeprimental conditions has not been tested in more than 2 animals and as such the statistics shown are without any value. At least 3 animals must be used/condition and data points for each animal must be shown.
It is unclear what figure 5 is about. Were there changes due to the treatments? Even if there were no changes representative images must be shown.
Figure 5-8. The nomenclature with ++ and - is confusing. Either proper quantification is performed or representative images shown side by side.
Figures 5-8. Instead of a magnification factor a scale bar should be shown.
Figures 5-8. Data from similar sized blood vessels should be shown.
Figure 8. VEGF staining in tissue is notoriously difficult. Is the antibody used well validated? Maybe these important findings could be validated by immunoblots, especially since the RNA did not change.
Discussion:
The authors make much of the fact that the ALDO concentration used is supra physiological. However, no comparative values are provided. It is also not clear what the local concentration in the skin is after application.
When comparing results with previous studies it is laudable that differences are pointed out. However, the description of published results should be more condensed (and in present tense). Instead better possible explanations underlying these differences should be offered and discussed.
Author Response
Comments and Suggestions for Authors
The authors investigate the effect of aldosterone in the rat skin. At relatively high concentrations aldosterone appeared to induce permeability, which was not associated with major overall alterations in protein expression. Involvement of MR and COX pathways was tested using pharmacological inhibitors.
The study results are potentially interesting. They do not agree much with published data (from other mainly in vitro systems). However, value of the presented data is vastly diminished by insufficient n numbers and the use of antibodies that have not been validated for IHC.
Specifically
Results:
All figures: I applaude the authors for being very precise about the number of animals used. However, what they claim to be n numbers are technical replicates. By my reckoning each exeprimental conditions has not been tested in more than 2 animals and as such the statistics shown are without any value. At least 3 animals must be used/condition and data points for each animal must be shown.
- Indeed, n for each group is the number of skin samples, not the number of animals, but we do not find them as replicates. For example – in the Fig. 2, we have 5 groups and 10 rats were used. The back of each of the rats was shaved and divided symmetrically into 4 areas. Different agents (ALDO 1-100 pmol, VEH, HIST) were randomly injected intradermally into those 4 skin areas (Figure 1A in the revised manuscript). So, from each rat 4 skin samples with different injected agent (different 4 groups) were obtained (in total we have 40 samples - 8 per group). Thus, each exeprimental condition has been tested in at least 8 animals, so the power of statistics was satisfied. We have included this explanation with corresponding figure in the revised manuscript. The protocol of our study is in agreement with the protocol from the original method [Udaka et al., Proc Soc Exp Biol Med 1970; Pietrzak et al., Clin Exp Dermatol 2009]. Moreover, this way of skin samples testing meets the criteria of 3R principles (Replacement, Reduction and Refinement) in animal experimentation.
Figure1A. Experimental protocol – intradermal injection of tested agents.
It is unclear what figure 5 is about. Were there changes due to the treatments? Even if there were no changes representative images must be shown.
- Figure 5 shows representative images of IHC staining for: vWF, VEGF, MR, HSD11β2, ZO-1 in the skin blood vessels under physiological conditions – it means in the skin samples from control group, without any treatment.
Figure 5-8. The nomenclature with ++ and - is confusing. Either proper quantification is performed or representative images shown side by side.
- A qualitative interpretation of IHC data is commonly used among scientists [Fedchenko et al., Diagn Pathol 2014]. In addition to the description of the evaluated parameters scientists use qualitative scoring systems to interpret received data, usually the force of IHC staining in different investigated areas. Score ranks usually lie in a range from “negative” (mostly marked as “-”) to “positive”, which may be signed with different amount of “+” depending on how many other categories lay between these border parameters. Most common spectrum of categories, describing different force of IHC expression in investigated groups, include: “negative”(−), “slight”(+), “moderate”(++), “strong”(+++) and their variations. One of the way of IHC data quantification is to present numbers or % of positive cells, although this is still the combination of quantitative and qualitative parameters. The scoring system could be:
“-“ ≤10 cells
“+” 10-≤30 cells
“++” 30-50 cells
“+++” ≥50 cells
However, the most common IHC data presentation, as a “gold standard” is the nomenclature with “++” and “-“, that we have used in our study (-/+).
Figures 5-8. Instead of a magnification factor a scale bar should be shown.
- As recommended, in the revised manuscript scale bars are shown on the Figures.
Figures 5-8. Data from similar sized blood vessels should be shown.
- We evaluated the microvessels in the rat skin, so the same type of vessels with similar size range, even if there were some differences in their diameter. This approach could be find in other studies as well [Zheng et al., Plast Reconstr Surg Glob Open 2015; Maurer et al, Ann Rheum Dis 2014]. Even if there are some differences in the vessel diameter, the strength of staining signal is not dependent on the vessel size. Figure 6 may confirm this thesis. The larger vessel (ALDO) showed no signal (-) compared to the smaller vessel (VEH) where the signal was strong (++).
Figure 8. VEGF staining in tissue is notoriously difficult. Is the antibody used well validated?
- We used antibody for VEGF staining that are commonly used, what makes it well validated. There are numerous references available in PubMed and provided by manufacturer as well confirming the quality of used VEGF antibody [Karamanolis et al., Mediators Inflamm 2013;https://www.labome.com/product/Dako/M7273.html]. We found strong staining for VEGF in the skin vessels with reproducible results. Moreover, no background staining for VEGF was observed, what makes the interpretation of results more reliable.
Maybe these important findings could be validated by immunoblots, especially since the RNA did not change.
- We are aware that the IHC staining is a qualitative method, although widely accepted. We agree with the Reviewer that these important results should be confirmed with some other method (e.g. immunoblotting), what we have already planned in our next study. The limitation of IHC staining we described at the end of the revised manuscript (Discussion p. 14; lines: 470-474).
Discussion:
The authors make much of the fact that the ALDO concentration used is supra physiological. However, no comparative values are provided. It is also not clear what the local concentration in the skin is after application.
- The physiological plasma aldosterone concentration goes from 55 to 250 pmol/L [Shidlovskyi et al., J Med Life 2019], so the doses of ALDO used in our study (1-100 pmol/0,1 ml 0,01 – 1 μmol/L) are considered as supraphysiological. However, even higher dose ranges have been used in many in vitro as well as in vivo studies investigating the mechanism of ALDO action. This is discussed in the revised manuscript as a limitation of the study (Discussion page 14, lines: 467-475). We did not measured the local concentration in the skin after ALDO application.
When comparing results with previous studies it is laudable that differences are pointed out. However, the description of published results should be more condensed (and in present tense). Instead better possible explanations underlying these differences should be offered and discussed.
- As recommended, we modified the discussion of the revised manuscript. The description of our results have been shortened and the differences with other studies are more pronounced in the revised discussion.
References:
- Udaka K, Takeuchi Y, Movat HZ. Simple method for quantitation of enhanced vascular permeability. Proc Soc Exp Biol Med 1970; 133: 1384–7.
- Pietrzak L, Mogielnicki A, Buczko W. Nicotinamide and its metabolite N- methylnicotina-mide increase skin vascular permeability in rats. Clin Exp Dermatol 2009; 34: 380–384.
- Fedchenko N, Reifenrath J. Different approaches for interpretation and reporting of immunohistochemistry analysis results in the bone tissue - a review. Diagn Pathol 2014; 9: 221.
- Zheng Z, Jian J, Velasco O, Hsu CY, Zhang K, Levin A, Murphy M, Zhang X, Ting K, Soo C. Fibromodulin Enhances Angiogenesis during Cutaneous Wound Healing. Plast Reconstr Surg Glob Open. 2015 Jan 8;2(12):e275. doi: 10.1097/GOX.0000000000000243.
- Maurer B, Distler A, Suliman YA, Gay RE, Michel BA, Gay S, Distler JH, Distler O. Vascular endothelial growth factor aggravates fibrosis and vasculopathy in experimental models of systemic sclerosis. Ann Rheum Dis. 2014 Oct;73(10):1880-7. doi: 10.1136/annrheumdis-2013-203535.
- Karamanolis G, Delladetsima I, Kouloulias V, Papaxoinis K, Panayiotides I, Haldeopoulos D, Triantafyllou K, Kelekis N, Ladas SD. Increased expression of VEGF and CD31 in postradiation rectal tissue: implications for radiation proctitis. Mediators Inflamm. 2013;2013:515048. doi: 10.1155/2013/515048.
- Shidlovskyi VO, Shidlovskyi OV, Tovkai OA, et al. Topical Diagnosis and Determination of the Primary Hyperaldosteronism Variant. J Med Life 2019; 12: 322–328.

Round 2
Reviewer 2 Report
The authors may well have conducted their animal experiments in a statistically relevant fashion. However, this is not borne out of what is described. Independent data points (these are the ones recorded from different animals which may represent averages from technical replicates must be shown (as requested previously) in addition or instead of the average value histograms.
I do not agree with the authors that the -, +, ++ annotation of images is something that is very common. This makes no sense in figure 5, as the strength of the signal may depend on the amount of antigen present or the quality of the antibody. As stated previously, Figure 5 does not provide any relevant data and should be suppressed. Instead control staining should be included in the experimentally relevant figures.
Similar vessels have to be shown when staining is compared. This is just the minimal standard accepted in vascular biology.
The dection of secreted VEGF protein is notoriously difficult and a contested subject in the angiogenesis field. If the authors can find a reference that shows validation of the antibody used in ko mice, this should be added. Otherwise, without any further evidence from e.g. RNA, it is indeed best to point out the limitations of relying on staining patterns from a single antibody.
Author Response
Comments and Suggestions for Authors
The authors may well have conducted their animal experiments in a statistically relevant fashion. However, this is not borne out of what is described. Independent data points (these are the ones recorded from different animals which may represent averages from technical replicates must be shown (as requested previously) in addition or instead of the average value histograms.
- As recommended, we replaced the average value histograms with scatter plots presenting independent data points. Now, Figures 2, 3, 4 and 8 (previous 9) show data presented as mean ± SEM and individual values as well.
I do not agree with the authors that the -, +, ++ annotation of images is something that is very common. This makes no sense in figure 5, as the strength of the signal may depend on the amount of antigen present or the quality of the antibody. As stated previously, Figure 5 does not provide any relevant data and should be suppressed. Instead control staining should be included in the experimentally relevant figures.
- As suggested, we removed Figure 5 from the revised manuscript. Although, we cannot include the control staining as we need new samples for this. We are not able to perform additional animal experiments at the moment.
Similar vessels have to be shown when staining is compared. This is just the minimal standard accepted in vascular biology.
- As requested, in the revised manuscript we showed data from blood vessels of similar size (now Figure 5 and Figure 6 – previously 6 and 7, renumbered after Figure 5 removal).
The detection of secreted VEGF protein is notoriously difficult and a contested subject in the angiogenesis field. If the authors can find a reference that shows validation of the antibody used in ko mice, this should be added. Otherwise, without any further evidence from e.g. RNA, it is indeed best to point out the limitations of relying on staining patterns from a single antibody.
- The antibody that we used for VEGF IHC staining (M7273; DakoCytomation, Denmark) was used previously by others in rats [Turan et al., Arch Gynecol Obstet, 2015; Ratajczak-Wielgomas et al., Folia Histochem Cytobiol, 2018] and mice [Veron et al., Kidney Int, 2010; Veron et al., Diabetologia, 2011; Favaron et al., PLoS One, 2014; Zibert et al., J Clin Invest, 2011; Holstein et a., Br J Pharmacol, 2008], and the IHC results were validated with other methods (ELISA, immunobloting). We added some of those references in the revised manuscript [61-65] and, as requested, we also pointed out the limitations of IHC relying on staining patterns from a single antibody at the end of the manuscript.
References:
- Turan GA, Eskicioglu F, Sivrikoz ON, Cengiz H, Adakan S, Gur EB, Tatar S, Sahin N, Yilmaz O. Myo-inositol is a promising treatment for the prevention of ovarian hyperstimulation syndrome (OHSS): an animal study. Arch Gynecol Obstet. 2015 Nov;292(5):1163-71. doi: 10.1007/s00404-015-3747-5.
- Ratajczak-Wielgomas K, Kassolik K, Grzegrzolka J, Halski T, Piotrowska A, Mieszala K, Wilk I, Podhorska-Okolow M, Dziegiel P, Andrzejewski W. Effects of massage on the expression of proangiogenic markers in rat skin. Folia Histochem Cytobiol. 2018;1(2):83-91. doi: 10.5603/FHC.a2018.0008.
- Veron D, Reidy KJ, Bertuccio C, Teichman J, Villegas G, Jimenez J, Shen W, Kopp JB, Thomas DB, Tufro A. Overexpression of VEGF-A in podocytes of adult mice causes glomerular disease. Kidney Int. 2010 Jun;77(11):989-99. doi: 10.1038/ki.2010.64.
- Veron D, Bertuccio CA, Marlier A, Reidy K, Garcia AM, Jimenez J, Velazquez H, Kashgarian M, Moeckel GW, Tufro A. Podocyte vascular endothelial growth factor (Vegf₁₆₄) overexpression causes severe nodular glomerulosclerosis in a mouse model of type 1 diabetes. Diabetologia. 2011 May;54(5):1227-41. doi: 10.1007/s00125-010-2034-z.
- Favaron PO, Mess A, Will SE, Maiorka PC, de Oliveira MF, Miglino MA. Yolk sac mesenchymal progenitor cells from New World mice (Necromys lasiurus) with multipotent differential potential. PLoS One. 2014 Jun 11;9(2):e95575. doi: 10.1371/journal.pone.0095575.
- Zibert JR, Wallbrecht K, Schön M, Mir LM, Jacobsen GK, Trochon-Joseph V, Bouquet C, Villadsen LS, Cadossi R, Skov L, Schön MP. Halting angiogenesis by non-viral somatic gene therapy alleviates psoriasis and murine psoriasiform skin lesions. J Clin Invest. 2011 Jan;121(1):410-21. doi: 10.1172/JCI41295.
- Holstein JH, Klein M, Garcia P, Histing T, Culemann U, Pizanis A, Laschke MW, Scheuer C, Meier C, Schorr H, Pohlemann T, Menger MD. Rapamycin affects early fracture healing in mice. Br J Pharmacol. 2008 Jul;154(5):1055-62. doi: 10.1038/bjp.2008.167.